# The Associations of Maternal Hemoglobin Concentration in Different Time Points and Its Changes during Pregnancy with Birth Weight Outcomes

**DOI:** 10.3390/nu14122542

**Published:** 2022-06-19

**Authors:** Zhicheng Peng, Shuting Si, Haoyue Cheng, Haibo Zhou, Peihan Chi, Minjia Mo, Yan Zhuang, Hui Liu, Yunxian Yu

**Affiliations:** 1Department of Public Health, and Department of Anesthesiology, Second Affiliated Hospital of Zhejiang University School of Medicine, Hangzhou 310058, China; 22018678@zju.edu.cn (Z.P.); 21818499@zju.edu.cn (S.S.); 3150101365@zju.edu.cn (H.C.); 11918158@zju.edu.cn (H.Z.); 22118872@zju.edu.cn (P.C.); minjiamo@zju.edu.cn (M.M.); yanzhuang@zju.edu.cn (Y.Z.); 2Department of Epidemiology & Health Statistics, School of Public Health, School of Medicine, Zhejiang University, Hangzhou 310058, China; 3Sir Run Run Shaw Hospital, School of Medicine, Zhejiang University, Hangzhou 310058, China; lhui2010@zju.edu.cn

**Keywords:** maternal hemoglobin, neonatal, birth weight, SGA/LBW, LGA/macrosomia, pregnancy

## Abstract

Maternal hemoglobin (Hb) is related to nutritional status, which affects neonatal birth weight. However, it is very common for maternal Hb to fluctuate during pregnancy. To evaluate the associations of maternal Hb in different time points and its changes during pregnancy with neonatal birth weight, small for gestational age (SGA)/low birth weight (LBW) and large for gestational age (LGA)/macrosomia, we conducted this study by using data from the Electronic Medical Record System (EMRS) database of Zhoushan Maternal and Child Care Hospital in Zhejiang province, China. The pregnancy was divided into five periods: first, early-second, mediate-second, late-second, early-third and late-third trimesters; we further calculated the maternal Hb changes during pregnancy. Overall, the socio-demographic characteristics, health-related information and childbirth-related information of 24,183 mother–infant pairs were obtained. The average Hb concentration during the different periods were 123.95 ± 10.14, 117.95 ± 9.84, 114.31 ± 9.03, 113.26 ± 8.82, 113.29 ± 8.68 and 115.01 ± 8.85 g/L, respectively. Significant dose–response relationships between maternal Hb and birth weight were observed in the first, late-second and later trimesters (*p* non-linear < 0.05). Maternal Hb < 100 g/L was related to a high risk of LGA/macrosomia in the late-second (OR: 1.47, 95% CI: 1.18, 1.83) and later trimesters; additionally, high maternal Hb (>140 g/L) increased the risk of SGA/LBW in the first (OR: 1.26, 95% CI: 1.01, 1.57) and late-third trimesters (OR: 1.96, 95% CI: 1.20, 3.18). In addition, the increase in maternal Hb from the late-second to late-third trimesters had a positive correlation with SGA/LBW. In conclusion, maternal Hb markedly fluctuated during pregnancy; the negative dose–response association of maternal Hb in the late-second and third trimesters, and Hb change during pregnancy with neonatal birth weight outcomes were observed, respectively. Furthermore, the phenomenon of high Hb in the first trimester and after the late-second trimester and the increase of maternal Hb from the late-second to late-third trimesters more significantly increasing the risk of SGA/LBW should especially be given more attention. Its biological mechanism needs to be further explored.

## 1. Introduction

Neonatal birth weight has been obtaining wide attention as it is a strong predictor of neonatal and perinatal mortality and disability, as well as birth weight percentiles, are used to predict the risk of growth disorders in newborns [1]. The global prevalence of low birth weight (LBW) was 14.6% in 2015 [2], and it was estimated that approximately 32.4 million infants that were small for gestational age (SGA) were born in low-income and middle-income countries in 2010 [3]. Both LBW and SGA were associated with short-term and long-term adverse outcomes, such as infection, respiratory depression, jaundice, obesity, insulin resistance and type 2 diabetes [1,4,5]. In addition, the prevalence of large for gestational age (LGA) and macrosomia is also increasing, especially in developing countries [6,7]. LGA and macrosomia are also considered factors that can increase the risk of early obesity, metabolic diseases [8], diabetes and early cardiovascular disease [9,10]. Growth restriction and excessive growth were two extremes of fetal development, both of which were associated with neonatal mortality and other adverse outcomes. Therefore, it is essential to identify the early factors that might influence neonatal birth weight and alert the need for an intervention.

Maternal nutritional status during pregnancy is one of the critical influencing factors of neonatal birth weight [11]. Both restricted growth and excessive growth of neonates were derived from altered metabolism in the uterus [12], which was closely related to maternal nutritional status. Hemoglobin (Hb) concentration plays an important role in maternal nutrition, especially the iron condition [13], and it can help to identify the neonatal adverse outcomes and alert the need for intervention measures as early as possible. Generally, hemoglobin testing is contained in routine blood examinations during pregnancy, and it is applied to screen for anemia in clinical practice [14]. However, the relationships of maternal Hb in different time points during pregnancy with fetal growth were controversial.

A retrospective analysis of Chinese pregnant women of Zhuang nationality reported a positive correlation between maternal Hb in the first trimester and birth weight and a negative correlation between Hb in the third trimester and birth weight [15]. Another study measured maternal Hb concentration at least once during pregnancy and found that the lowest maternal Hb during pregnancy might have an inverted U-shaped relationship with neonatal birth weight [16]. Furthermore, the results of meta-analyses suggested that in low-income countries, 25% of low birth weight (LBW) can be attributed to anemia (Hb < 110 g/L) [17]. Some other studies found that maternal Hb < 110 g/L in the first trimester was associated with neonatal adverse outcomes, including LBW, SGA and neonatal mortality [18,19], though no significant associations were observed in the second and third trimesters [19,20]. Nevertheless, another retrospective cohort analysis of 173,031 pregnant women found that anemia in the first and second trimesters was not associated with SGA [21]. In addition, a study found a faint U-shaped relationship between maternal Hb concentration at ≤20 weeks of gestational age and risk of neonatal adverse outcomes (including preterm deliveries, SGA) [22]. However, another study from Northwest China showed a U-shaped relationship between Hb concentration in the third trimester and the risk of SGA/LBW, as well as an inverted U-shaped relationship between maternal Hb and neonatal birth weight [23]. Meanwhile, several studies reported the association between Hb reduction and birth outcomes [24,25]. However, the definition of gestational age for maternal Hb measurements was not precise in all studies, which was considered a confounding factor affecting the association between maternal Hb and birth outcomes [26]. The maternal plasma volume generally expands from 10 weeks of gestational age and then reaches the peak at about 34–36 gestational weeks; after that, it keeps the peak or slowly returns to the level found at 10 weeks of gestational age [27]. Although the volume of red blood cells also increases during pregnancy, the increased plasma volume is larger, and then results in a decline in Hb concentration during pregnancy [25]. In addition, the ability of the placenta to adapt the uterine nutrition, which was thought to maximize fetal growth and viability at birth, also depended on the gestational timing [28,29].

However, whether there was a different effect of Hb measured at the different periods on the birth weight and the association of maternal Hb concentration changes during pregnancy with birth weight outcomes were unclear due to the fluctuation of maternal Hb concentration and change in placental adaptability during pregnancy. The present study defined five time points for maternal Hb measurement and aimed to explore the associations between maternal Hb concentrations in different time points during pregnancy and neonatal birth weight outcomes (including birth weight, SGA/LBW and LGA/macrosomia). Moreover, the effects of maternal Hb concentration changes during pregnancy on birth weight outcomes were also assessed.

## 2. Materials and Methods

### 2.1. Data Sources

The data were obtained from a comprehensive Electronic Medical Record System (EMRS) in Zhoushan Maternal and Child Care Hospital, which is a non-private-sector hospital. The EMRS was a municipal system and was established in Zhoushan in 2001 and included a prenatal health dataset and birth registration information dataset. In total, 78,297 pregnant women from January 2001 to May 2018 were recorded in the EMRS. From 2001 to 2009, the EMRS only included the data of the Zhoushan Maternal and Child Care Hospital, and after 2010, it contained the data of all the maternal and children’s health care in Zhoushan city. However, the routine blood data, which included maternal Hb levels, was only available in the data from Zhoushan Maternal and Child Care Hospital.

Maternal information about socio-demographic characteristics (e.g., maternal age, educational level, parity, last menstrual period, follow-up date) and health-related characteristics (e.g., maternal weight and height; laboratory parameters, such as Hb and glucose levels; systolic and diastolic blood pressure during pregnancy; liver and kidney disease) were extracted from the prenatal health dataset and birth information (e.g., neonatal gender, birth weight, gestational age at delivery, mode of delivery) of newborns was extracted from the birth registration information dataset. A unique personal identification number was provided to link both datasets. The study protocol was approved by the institutional review board of Zhejiang University School of Medicine.

### 2.2. Study Population

The pregnant women received their first prenatal check-up before 14 weeks of gestational age. Then, they attended check-ups every four weeks in the first and second trimesters, and then every two weeks before 37 weeks of gestational age according to the guidelines. Pregnant women who met the following criteria were included in the study: (1) maternal age > 18 years old, (2) singleton pregnancy, (3) pregnant women with at least one record of hemoglobin concentration during pregnancy and (4) pregnant women who delivered their baby after 32 weeks of gestational age. The exclusion criteria included (1) women with hypertensive disorders or gestational diabetes mellitus (GDM), (2) maternal BMI in the first trimester of >40 kg/m^2^ or <15 kg/m^2^ and (3) missing data for maternal Hb during pregnancy or neonatal birth weight.

### 2.3. Definition of Exposure

The maternal Hb concentration was measured through routine blood tests during pregnancy. Due to fluctuations in the maternal Hb concentration during pregnancy, the gestational period was divided into six time periods to explore the effects of Hb concentration in different time points during pregnancy on the birth outcomes: first (5–13 weeks of gestational age), early-second (14–17 weeks of gestational age), mediate-second (18–22 weeks of gestational age), late-second (23–27 weeks of gestational age), early-third (28–31 weeks of gestational age) and late-third trimesters (from 32 weeks of gestational age to the time of delivery). Meanwhile, the mean Hb concentrations around 100 and 140 g/L were classified into 6 categories using intervals of 10 g/L: <100 g/L, 100~109 g/L, 110~119 g/L, 120~129 g/L, 130~139 g/L and ≥140 g/L. A similar classification was also reported in some other studies [30,31].

The maternal Hb concentration reaches a minimum in the late-second to early-third trimesters and then rises in the late-third trimester [32,33]. When we evaluated the changes in Hb concentration during pregnancy, the Hb concentrations in the late-second trimester and late-third trimester were chosen, which might appropriately reflect the changes in maternal Hb during pregnancy. Therefore, the changes in Hb concentration from the first to late-second trimesters, from the first to late-third trimesters and from the late-second to late-third trimesters were evaluated using Hbfs (Hb in the late-second minus Hb in first), Hbft (Hb in the late-third minus Hb in first) and Hbst (Hb in the late-third minus Hb in late-second). Furthermore, Hbfs, Hbft and Hbst were divided into four categories based on their quartiles.

### 2.4. Definition of Outcome

The data on birth outcomes were obtained from the birth registration information datasets. SGA and LGA were defined as the top and bottom ten percent of gestational age- and gender-specific birth weights, respectively. LBW and macrosomia were defined as infant birth weight <2500 g and ≥4000 g, respectively [7]. The compound outcomes were defined as the following: the infants with SGA or LBW were grouped as “Lightweight” and those with LGA or macrosomia were grouped as “Heavyweight”; otherwise, they were grouped as “Normalweight”. Furthermore, infants with “Lightweight” and “Heavyweight” were grouped as “TotalAdverse”.

### 2.5. Statistical Analysis

Continuous variables are presented as mean ± standard deviation (SD), while categorical parameters are described as number (N) and percentage (%). The characteristics of study participants in the Lightweight, Heavyweight and Normalweight groups were compared using the analysis of variance (ANOVA) for continuous variables and the chi-square test for categorical variables. A restricted cubic spline (RCS) function with 5 knots was applied to analyze the association of maternal Hb concentration in different time points and its changes during pregnancy with neonatal birth weight outcomes [34]. Multinomial logistic regression models were used to analyze the associations of Hb concentration in different time points and Hb changes during pregnancy with neonatal compound outcomes. The maternal Hb concentration at 110~119 g/L was set as the reference group in each time point and the first quartile (Q1) in Hb change was set as the reference group. The odds ratios (Ors) and 95% confidence intervals (Cis) for “Lightweight” and “Heavyweight” were used to evaluate the relative risk. The covariants included maternal age, maternal education, parity, gestational age of Hb measurement, neonatal gender, gestational age at delivery and mode of delivery; they were adjusted in the above analysis model 1. Besides the covariants in model 1, model 2 was additionally adjusted for maternal early pregnancy BMI and weight gain during pregnancy. Sensitivity analysis was conducted among pregnant women with an early pregnancy BMI < 24 kg/m^2^ and pregnant women who delivered their babies after 37 weeks of gestational age. All the statistical analyses were performed in R software (version 4.0.3) and *p* < 0.05 was considered statistically significant.

## 3. Results

### 3.1. Population Characteristics

A total of 24,183 mother–infant pairs were included in the final analysis (Appendix A) and the comparison of general information between participants and non-participants is shown in Appendix A. The maternal socio-demographic, health-related characteristics and neonatal birth characteristics stratified by the compound outcomes are shown in Table 1. The Lightweight and Heavyweight groups accounted for 8.5% and 11.2%, respectively. There were significant differences in maternal age; maternal education level; primipara; BMI at the first trimester; weight gain; and all neonatal characteristics between the Normalweight, Lightweight and Heavyweight groups.

### 3.2. Maternal Hb Status in Different Time Points and Hb Changes during Pregnancy

In the first, early-second, mediate-second, late-second, early-third and late-third trimesters, the averages of the maternal Hb concentrations were 123.95 ± 10.14, 117.95 ± 9.84, 114.31 ± 9.03, 113.26 ± 8.82, 113.29 ± 8.68 and 115.01 ± 8.85 g/L, respectively; the proportions of maternal Hb concentrations <110 g/L were 6.8%, 17.5%, 28.8%, 32.7%, 33.4% and 27.3%, respectively. Before the late-second trimester, there was no significant difference in Hb level between the Normalweight, Lightweight and Heavyweight groups, except in the first trimester. However, in the late-second and third trimesters, there were significant differences in the Hb levels between the Normalweight, Lightweight and Heavyweight groups (Figure 1). In addition, the mean values of Hbfs, Hbft and Hbst were significantly different between the Normalweight, Lightweight and Heavyweight groups. The details are presented in Appendix A.

### 3.3. Association of Maternal Hb Level and Its Changes during Pregnancy with Neonatal Birth Weight Outcomes

Multivariate RCS models were used to assess the non-linear relationships of maternal Hb in different time points of pregnancy and its changes during pregnancy with birth outcomes. The non-linear associations of maternal Hb in the early-second and mediate-second trimester with neonatal birth weight were not observed (Appendix A). Meanwhile, the overall association and non-linear relationships between maternal Hb and birth weight were found in the first, late-second, early-third and late-third trimesters (*P*_non-linear_ were 0.011, 0.009, <0.001 and <0.001, respectively). An inverted U-shaped association between maternal Hb and birth weight was observed in the first trimester. In addition, the decrease in neonatal weight was steeper in the 105–120 g/L range of maternal Hb and a roughly negative correlation of maternal with neonatal birth weight was observed in the late-second and later trimesters (Figure 2).

For the compound outcomes, non-linear relationships of maternal Hb in the first, early-second, mediate-second and late-second trimesters with Lightweight, Heavyweight and TotalAdverse were not observed. (Appendix A). In the early-third trimester, significant dose–response associations of maternal Hb with Lightweight and Heavyweight were observed but were not observed in the association of maternal Hb with TotalAdverse. In the late-third trimester, maternal Hb had a significant U-shaped association with Lightweight, and the corresponding maternal Hb where the risk of Lightweight was lowest was about 110 g/L; moreover, when maternal Hb > 120 g/L, the risk of TotalAdverse significantly increased (Figure 3).

In addition, Table 2 displays the associations between maternal Hb concentration in different time points and neonatal compound outcomes. In model 2, compared with women with Hb = 110~119 g/L, those with Hb < 100 g/L had a significantly decreased risk of Lightweight only in the early-third trimester (OR: 0.66, 95% CI: 0.49, 0.88) and those with Hb > 140 g/L had a significantly increased risk of Lightweight only in the first trimester (OR: 1.26, 95% CI: 1.01, 1.57) and late-third trimester (OR: 1.96, 95% CI: 1.20, 3.18); moreover, women with Hb > 120 g/L had a trend of increased risk of Lightweight in the late-third trimester (*P*_trend_ < 0.001). In the mediate-second trimester, compared with women with Hb = 110~119 g/L, pregnant women with Hb = 100~109 g/L had a significantly increased risk of Heavyweight (OR: 1.12, 95% CI: 1.01, 1.25). Compared with women with Hb = 110~119 g/L, women with Hb < 100 had an increased risk of Heavyweight at the late-second and later trimester (late-second trimester: OR: 1.47, 95% CI: 1.18, 1.83; early-third trimester: OR: 1.30, 95% CI: 1.04, 1.63; late-third trimester: OR: 1.38, 95% CI: 1.09, 1.76).

On the other hand, there was a positive correlation between maternal Hb reduction during pregnancy and neonatal birth weight, where a non-linear dose–response relationship was found between Hbst and neonatal birth weight (*P*_non-linear_ was <0.001) (Figure 4), but not found between Hbfs, Hbft and Hbst and the compound outcomes (Appendix A). A negative association of the decline in maternal Hb with the risk of Lightweight was found in the first to late-second trimesters, first to late-third trimesters and late-second to late-third trimesters; however, a significant association between the decline in Hb and Heavyweight was not observed in the late-second to late-third trimesters. Compared with pregnant women with Hbfs in Q1, those with Hbfs in Q2 had a significantly decreased risk of Heavyweight (OR: 0.83, 95% CI: 0.73, 0.94). In addition, as the maternal Hb decreased more during pregnancy, the risk of Lightweight was lower (all *P*_trend_ < 0.001) (Table 3). Similar results were observed in pregnant women with early-pregnancy BMI < 24 kg/m^2^, pregnant women who delivered their baby after 37 weeks of gestational age and pregnant women who had at least one record of Hb concentration during each defined period (Appendix A).

## 4. Discussion

This study found a faint inverted U-shaped association between maternal Hb in the first trimester and birth weight, and significant negative associations between maternal Hb and birth weight outcomes (including birth weight, Lightweight and Heavyweight) were mainly observed in the late-second and later trimester. Furthermore, the maternal Hb decline during pregnancy was positively associated with neonatal birth weight and negatively associated with the risk of Lightweight.

In this study, an inverted U-shaped association between maternal Hb in the first trimester and neonatal birth weight was observed. Both extremely low and extremely high maternal Hb in the first trimester might be associated with LBW/SGA. However, many studies found a significant positive correlation between maternal Hb in the first trimester and fetal birth weight, as well as a significant association between low maternal Hb and SGA/LBW [26,35,36]. Other studies did not find a significant association between maternal Hb in the first trimester and birth weight outcomes. A cross-sectional study reported that maternal Hb in the first trimester was interpreted as having little effect on birth weight [37]. Another meta-analysis found that anemia (Hb < 110 g/L) diagnosed before 20 weeks of gestational age did not increase the risk of LBW and SGA [38]; similar results were also shown in a study in Suzhou, China [33]. The associations of maternal Hb in the first trimester with neonatal birth weight outcomes were inconsistent, possibly due to the heterogeneity of a different study population. Another reason for the inconsistency might be unclear iron status and iron supplementation during pregnancy, which is also associated with birth weight outcomes [39].

In addition, the associations of maternal Hb < 100 g/L in the early- and mediate-second trimester with neonatal birth weight and Lightweight were not observed. Similar to our results, some other studies also reported that the associations of maternal Hb in the second trimester with fetal birth weight and LBW were insignificant [26,36]. Some studies reported that severe anemia may cause poor fetal growth in utero because of insufficient oxygen flow to the placental tissue [26], and ultimately, affect fetal birth weight and cause LBW [40]. However, because of the deficient data with few pregnant women Hb < 70 g/L in this study, we were unable to assess the association between severe anemia (Hb < 70 g/L) and birth outcomes. Moreover, in the late-second and later trimester, maternal Hb < 110 g/L was related to the increased birth weight and increased risk of Heavyweight; a review also indicated that low maternal Hb in the late-second trimester was associated with a high risk of LGA [41], which might be due to placental hypertrophy [42]. Additionally, Scanlon et.al reported that a high Hb (>144 g/L) level in the second trimester was associated with SGA [21], which was different from our result showing that maternal Hb > 140 g/L in the second trimester was not related to Lightweight or Heavyweight. This may have been due to the small sample of pregnant women with Hb > 140 g/L in the second trimester in this study.

Maternal Hb concentration in the late-third trimester had a roughly negative association with neonatal birth weight and a faint U-shaped association with Lightweight. When maternal Hb was 100–110 g/L, the risk of Lightweight was the lowest and the risk of Heavyweight was relatively high. Similar to our results, Chen et al. showed that maternal Hb in the third trimester was inversely correlated with neonatal birth weight [15]. Relatively low maternal Hb in the third trimester usually reflects changes in plasma volume rather than poor maternal nutrition or adaptation [43]. However, a recent prospective study from Northwest China showed an inverted U-shaped association between maternal Hb in the third trimester and neonatal birth weight [23], and a significant positive correlation between maternal Hb in the third trimester and birth weight was reported in another study [44]. This may be because, even in the third trimester, maternal Hb in different periods reflects different conditions of the intrauterine environment. As presented in a study, maternal Hb < 110 g/L was related to an increased placental ratio but whether it reflected the placental hypertrophy or restriction of fetal growth was unclear [45]. More studies are needed to explore this mechanism. Maternal high Hb was mainly related to the risk of Lightweight in the late-third trimester. A study from Norway also reported that increased Hb levels were associated with lower placental weight and impaired fetal growth [46]. Abnormally high Hb concentrations during pregnancy usually indicate poor plasma volume expansion, which also leads to an increased risk of LBW [20,21,41,47]. In addition, high Hb might increase blood flow resistance, which reduces maternal blood perfusion and leads to placental dysplasia [48].

On the other hand, we found that the decrease in maternal Hb from first to late-second trimester and from first to late-third trimester was positively correlated with birth weight and negatively correlated with the risk of Lightweight. Similar to our results, preceding studies showed that changes in Hb concentration during pregnancy were associated with adverse birth outcomes. Rasmussen et al. found that a decrease in maternal Hb from the first to second trimester had a strong positive correlation with neonatal birth weight [37]. The same relationship was also observed in the first to third trimesters in other studies [15,24]. Pregnant women with the lowest Hb reduction from early to late pregnancy had an increased risk of LBW and SGA compared with women with an intermediate Hb reduction [15,24]. Changes in Hb during pregnancy were significantly correlated with changes in plasma volume [43,49]. A smaller decrease in maternal Hb from the first trimester may indicate a reduction in plasma volume expansion, which may impair fetoplacental circulation and increase the risk of LBW and SGA [41]. Another possible mechanism was that expansion of the plasma volume may reduce blood viscosity and favor blood flow in the maternal intervillous space [50]. Nevertheless, the positive associations of maternal Hb decline from the first- to late-second trimesters and from the first- to late-third trimesters with Heavyweight were found; we speculated that a larger reduction in Hb levels may indicate more erythropoietin secretion, which was associated with angiogenesis [51]. More studies are necessary to explore this mechanism. The decrease in maternal Hb levels from the late-second to late-third trimesters was related to decreased risk of Lightweight, but not related to Heavyweight. Fetal growth reached a peak in the late-third trimester and the nutrients required by the fetus increased dramatically [52]. Therefore, the reduction in Hb, which implied an increased expansion of plasma volume, may accelerate the transport of nutrients in the body of pregnant women. As far as we know, no study has reported similar results. The above results were shown in pregnant women with early-pregnancy BMI < 24 kg/m^2^ or pregnant women who delivered their baby after 37 weeks of gestational age in a sensitivity analysis.

The results of this study showed that the associations of maternal Hb with neonatal birth weight outcomes were different due to the timing of maternal Hb measurements. In addition, maternal Hb change during pregnancy was also strongly associated with birth weight outcomes. Maternal Hb > 120 g/L in the late-third trimester and an increase in maternal Hb from the late-second to late-third trimesters should especially be given more attention. The maternal Hb fluctuated during pregnancy; therefore, the findings in this study were important for clinically identifying the associations of maternal Hb at different time points and its changes during pregnancy with neonatal birth weight outcomes, which could provide recommendations for optimal interventions. Moreover, the research findings might contribute to clinical evidence-based studies to determine the pathological value of maternal Hb in different periods during pregnancy. The association of maternal Hb in different time points with neonatal birth weight outcomes might be partly explained by placental adaptation. Further study should be conducted to explore more biological mechanisms.

## 5. Strengths and Limitations

There were several strengths to this study. Relatively drastic physiological changes in maternal Hb occurred with the progress of pregnancy; accordingly, even in the second or third trimester, different time points for Hb measurements may lead to heterogeneity in results. We further divided the second and third trimesters into five time periods and explored the relationships between Hb concentrations in different periods and birth weight outcomes. Moreover, we selected the value of Hb concentration in a specific period to calculate the maternal Hb changes during pregnancy, which were more representative. We also collected data on maternal early-pregnancy BMI and weight gain during pregnancy, which were recognized as important factors that had a greater impact on birth weight.

Some limitations to this study should also be taken into consideration. First, the proportion of pregnant women with severe anemia (Hb < 70 g/L) in our data was very low; therefore, we were unable to verify that maternal Hb < 70 g/L was associated with adverse birth weight outcomes [53,54], and the generalizability of the research findings might be limited due to the significant differences between participants and non-participants; therefore, several subgroup analysis and sensitivity analysis were conducted to examine the stability of the results. Second, the information on iron supplementation for pregnant women was mostly missing. The association between iron supplementation and maternal Hb may have had an influence on this study. The relationship between changes in Hb levels and birth weight outcomes was evaluated to diminish the impact. Third, whether maternal low Hb was caused by iron deficiency was unclear. However, iron deficiency is the most common cause of anemia, especially in developing countries [35]; therefore, we can reasonably assume that the majority of pregnant women with low Hb were suffering from iron deficiency. In addition, information on complications such as placental insufficiency was not available, which might affect the association between maternal Hb and neonatal birth weight; further study is needed to eliminate the effect of complications.

## 6. Conclusions

Maternal Hb markedly fluctuated during pregnancy; a negative dose–response association of maternal Hb in the late-second and third trimesters and a Hb change during pregnancy with neonatal birth weight outcomes were observed. Furthermore, the phenomena of high Hb in the first trimester and after the late-second trimester and the increase in maternal Hb from the late-second to late-third trimesters more significantly increasing the risk of SGA/LBW should especially be given more attention. Its biological mechanism needs to be further explored.

## Figures and Tables

**Figure 1 nutrients-14-02542-f001:**
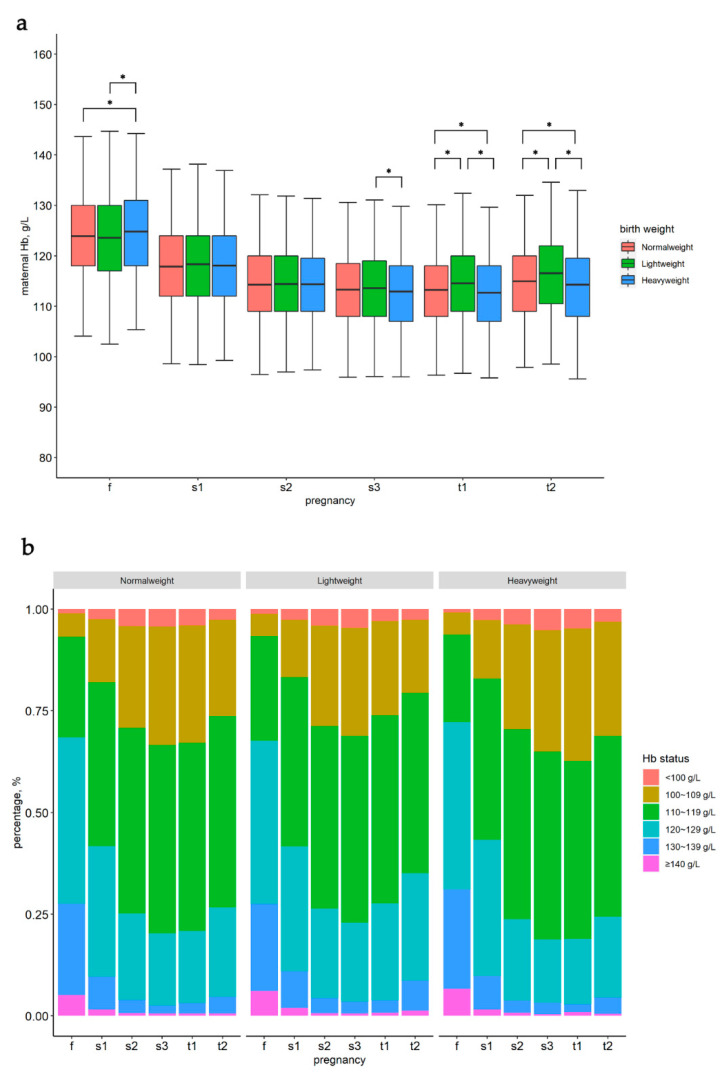
Maternal hemoglobin concentration (**a**) and percentage of maternal hemoglobin status (**b**) stratified by birth weight outcomes at different pregnancy time points. f, first trimester; s1, early-second trimester; s2, mediate-second trimester; s3, late-second trimester; t1, early-third trimester; t2, late-third trimester; * significant difference between the groups.

**Figure 2 nutrients-14-02542-f002:**
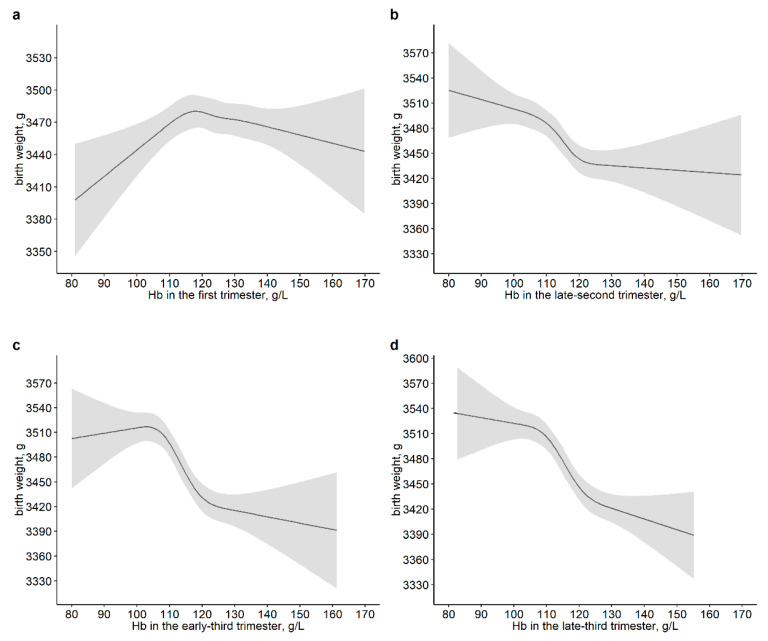
Dose–response associations of maternal hemoglobin in the first trimester (**a**), in the late-second trimester (**b**), in the early-third trimester (**c**) and in the late-third trimester (**d**) with neonatal birth weight. The *p*-values for overall associations were <0.001. The *p*-values for non-linear associations were 0.011 (**a**), 0.009 (**b**), <0.001 (**c**) and <0.001 (**d**), respectively. Models were adjusted for maternal age, education, parity, gestational age of hemoglobin measurement, neonatal gender, gestational age at delivery, mode of delivery, maternal BMI in the first trimester and weight gain during pregnancy.

**Figure 3 nutrients-14-02542-f003:**
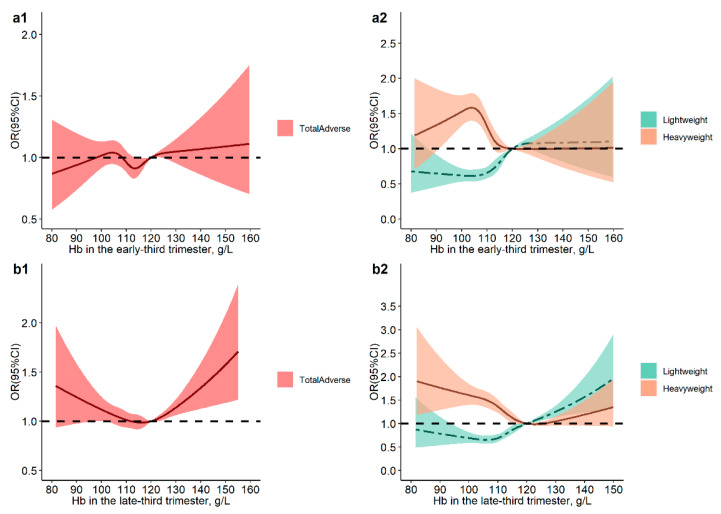
Dose–response associations of maternal hemoglobin in the early-third trimester (**a**) and in the late-third trimester (**b**) with neonatal complex outcomes (including SGA/LBW, LGA/macrosomia and total birth weight adverse outcomes). The *p*-values for overall associations were <0.001. The *p*-values for non-linear associations between Hb and total adverse outcomes were 0.064 (**a1**) and 0.002 (**b1**). The *p*-values for non-linear associations between Hb and SGA/LBW were 0.003 (**a2**) and 0.028 (**b2**), while the *p*-values for non-linear associations between Hb and LGA/macrosomia were <0.001 (**a2**) and <0.001 (**b2**). The models were adjusted for maternal age, education, parity, gestational age of hemoglobin measurement, neonatal gender, gestational age at delivery, mode of delivery, maternal BMI in the first trimester and weight gain during pregnancy. Hemoglobin at 120 g/L was set as the reference level.

**Figure 4 nutrients-14-02542-f004:**
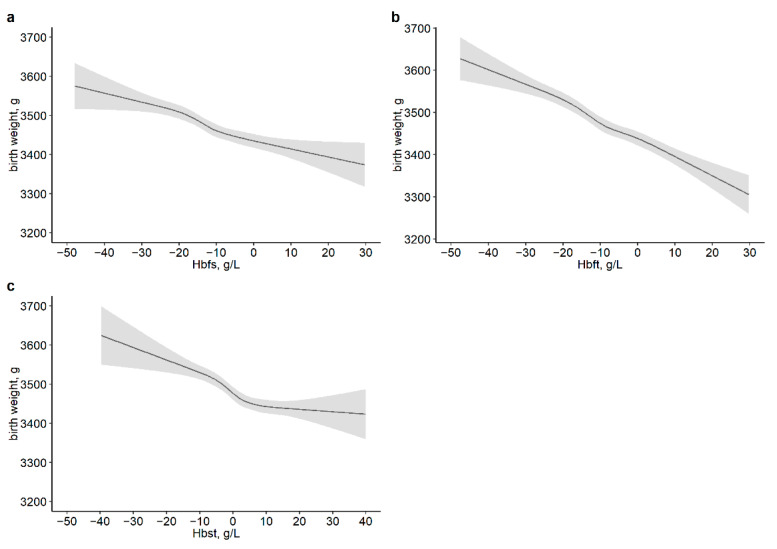
Dose–response associations of maternal hemoglobin change from first to late-second trimesters (**a**), from first to late-third trimesters (**b**) and from late-second to late-third trimesters (**c**) with neonatal birth weight. The *p*-values for overall associations were <0.001. The *p*-values for non-linear associations were 0.114 (**a**), 0.390 (**b**) and <0.001 (**c**). Models were adjusted for maternal age, education, parity, neonatal gender, gestational age at delivery, mode of delivery, maternal BMI in the first trimester, weight gain during pregnancy, and Hb concentration in the first trimester or in the late-second trimester. Hbfs, Hb in late-second minus Hb in first; Hbft, Hb in late-third minus Hb in first; Hbst, Hb in late-third minus Hb in late-second.

**Table 1 nutrients-14-02542-t001:** Comparison of maternal and neonatal basic characteristics between groups.

Variables	Normalweight	Lightweight	Heavyweight	*p*
	N = 19,413	N = 2055	N = 2715	
**Maternal Characteristics**				
Maternal age, years, mean ± SD	26.62 ± 3.70	26.49 ± 3.59	26.91 ± 3.94	<0.001
Education, N (%)				
Primary school or below	335 (1.7)	51 (2.5)	55 (2.1)	0.006
Junior high school	6603 (34.3)	682 (33.4)	985 (36.7)	
Senior high school	4915 (25.6)	488 (23.9)	671 (25.0)	
College or above	7383 (38.4)	820 (40.2)	971 (36.2)	
Primipara, N (%)				
Yes	11,622 (59.9)	1221 (59.4)	1638 (60.3)	<0.001
No	1855 (9.6)	138 (6.7)	367 (13.5)	
Unknown	5936 (30.6)	696 (33.9)	710 (26.2)	
BMI at 1st trimester, kg/m^2^, mean ± SD	20.58 ± 2.53	19.95 ± 2.48	21.83 ± 2.85	<0.001
BMI categories, N (%)				
Underweight (<18.5)	13,820 (71.2)	1295 (63.0)	1928 (71.0)	<0.001
Normalweight (18.5–23.9)	3878 (20.0)	616 (30.0)	242 (8.9)	
Overweight (24.0–27.9)	1458 (7.5)	128 (6.2)	462 (17.0)	
Obesity (≥28.0)	257 (1.3)	16 (0.8)	83 (3.1)	
WG * during pregnancy, kg, mean ± SD	13.88 ± 3.79	12.61 ± 3.95	15.41 ± 4.06	<0.001
**Neonatal Characteristics**				
Gender of newborn, N (%)				
Male	10,208 (52.6)	878 (42.7)	1769 (65.2)	<0.001
Female	9205 (47.4)	1177 (57.3)	946 (34.8)	
Mode of delivery, N (%)				
Vaginal	7838 (40.4)	956 (46.5)	599 (22.1)	<0.001
Cesarean	10,515 (54.2)	995 (48.4)	1953 (71.9)	
Forceps	205 (1.1)	30 (1.5)	25 (0.9)	
Others	855 (4.4)	74 (3.6)	138 (5.1)	
GA * at delivery, weeks, mean ± SD	39.15 ± 1.17	38.59 ± 2.07	39.34 ± 1.35	<0.001
Birth weight, g, mean ± SD	3.38 ± 0.29	2.64 ± 0.31	4.14 ± 0.31	<0.001
Preterm, N (%)				
No	19,048 (98.1)	1760 (85.6)	2637 (97.1)	<0.001
Yes	365 (1.9)	295 (14.4)	78 (2.9)	

Lightweight, small for gestational age (SGA) or low birth weight (LBW); Heavyweight, large for gestational age (LGA) or macrosomia; * WG, weight gain; GA, gestational age.

**Table 2 nutrients-14-02542-t002:** Associations between maternal hemoglobin concentration at different periods and adverse birth outcomes.

Hemoglobin	Normalweight	Lightweight			Heavyweight		
	N (%)	N (%)	Model 1 †	Model 2 ‡	N (%)	Model 1 †	Model 2 ‡
			OR (95% CI)	OR (95% CI)		OR (95% CI)	OR (95% CI)
First trimester							
<100	189 (1.0)	30 (1.5)	1.51 (1.01, 2.25)	1.45 (0.96, 2.18)	17 (0.6)	0.72 (0.44, 1.20)	0.73 (0.43, 1.23)
100~109	1136 (5.9)	136 (6.6)	1.15 (0.93, 1.40)	1.11 (0.91, 1.37)	139 (5.1)	0.97 (0.79, 1.18)	1.03 (0.84, 1.26)
110~119	4830 (24.9)	501 (24.4)	Ref.	Ref.	613 (22.6)	Ref.	Ref.
120~129	7911 (40.8)	821 (40.0)	1.00 (0.89, 1.12)	1.03 (0.91, 1.16)	1115 (41.1)	1.10 (0.99, 1.23)	1.02 (0.92, 1.14)
130~139	4356 (22.4)	447 (21.8)	1.00 (0.87, 1.14)	1.06 (0.92, 1.22)	665 (24.5)	1.17 (1.04, 1.32)	1.05 (0.93, 1.19)
≥140	991 (5.1)	120 (5.8)	1.17 (0.95, 1.46)	1.26 (1.01, 1.57)	166 (6.1)	1.27 (1.05, 1.54)	1.04 (0.86, 1.27)
Early-second trimester							
<100	381 (2.4)	43 (2.6)	1.00 (0.72, 1.40)	0.93 (0.67, 1.31)	60 (2.7)	1.20 (0.90, 1.60)	1.34 (0.99, 1.80)
100~109	2445 (15.3)	230 (13.9)	0.90 (0.76, 1.05)	0.86 (0.73, 1.01)	328 (14.7)	1.01 (0.88, 1.15)	1.12 (0.97, 1.29)
110~119	6407 (40.2)	672 (40.5)	Ref.	Ref.	869 (39.0)	Ref.	Ref.
120~129	5119 (32.1)	533 (32.1)	0.99 (0.88, 1.12)	1.05 (0.93, 1.19)	752 (33.7)	1.06 (0.95, 1.18)	0.96 (0.86, 1.07)
130~139	1323 (8.3)	147 (8.9)	1.08 (0.89, 1.31)	1.20 (0.99, 1.46)	182 (8.2)	0.97 (0.82, 1.16)	0.80 (0.67, 0.96)
≥140	266 (1.7)	33 (2.0)	1.13 (0.77, 1.65)	1.25 (0.85, 1.83)	38 (1.7)	1.07 (0.75, 1.53)	0.87 (0.60, 1.26)
Mediate-second trimester							
<100	694 (4.0)	73 (4.0)	0.98 (0.76, 1.27)	0.87 (0.67, 1.13)	83 (3.4)	0.86 (0.68, 1.10)	1.04 (0.81, 1.33)
100~109	4268 (24.8)	450 (24.5)	1.02 (0.90, 1.15)	0.96 (0.85, 1.09)	608 (25.1)	1.00 (0.90, 1.12)	1.12 (1.00, 1.25)
110~119	7865 (45.6)	815 (44.3)	Ref.	Ref.	1130 (46.7)	Ref.	Ref.
120~129	3720 (21.6)	428 (23.3)	1.10 (0.97, 1.25)	1.17 (1.03, 1.33)	500 (20.7)	0.93 (0.83, 1.04)	0.82 (0.73, 0.93)
130~139	588 (3.4)	61 (3.3)	1.04 (0.79, 1.38)	1.11 (0.83, 1.46)	81 (3.3)	0.92 (0.72, 1.18)	0.79 (0.61, 1.02)
≥140	104 (0.6)	11 (0.6)	0.95 (0.50, 1.80)	1.04 (0.54, 2.00)	16 (0.7)	1.10 (0.64, 1.89)	0.95 (0.55, 1.66)
Late-second trimester							
<100	714 (4.1)	83 (4.6)	1.11 (0.87, 1.42)	0.99 (0.78, 1.27)	111 (4.6)	1.23 (0.99, 1.52)	1.47 (1.18, 1.83)
100~109	4885 (28.4)	497 (27.3)	0.99 (0.88, 1.12)	0.93 (0.82, 1.05)	733 (30.5)	1.15 (1.03, 1.27)	1.29 (1.16, 1.43)
110~119	8036 (46.7)	820 (45.0)	Ref.	Ref.	1068 (44.5)	Ref.	Ref.
120~129	3097 (18.0)	351 (19.3)	1.12 (0.98, 1.28)	1.21 (1.06, 1.39)	412 (17.2)	0.98 (0.86, 1.11)	0.87 (0.76, 0.99)
130~139	395 (2.3)	60 (3.3)	1.49 (1.11, 1.98)	1.59 (1.19, 2.14)	67 (2.8)	1.30 (0.99, 1.71)	1.08 (0.81, 1.43)
≥140	83 (0.5)	10 (0.5)	1.05 (0.53, 2.07)	1.19 (0.59, 2.37)	10 (0.4)	1.01 (0.52, 1.98)	0.89 (0.45, 1.76)
Early-third trimester							
<100	716 (4.2)	58 (3.2)	0.73 (0.55, 0.97)	0.66 (0.49, 0.88)	104 (4.4)	1.12 (0.90, 1.40)	1.30 (1.04, 1.63)
100~109	4929 (29.2)	442 (24.5)	0.84 (0.74, 0.95)	0.80 (0.70, 0.90)	794 (33.3)	1.24 (1.12, 1.37)	1.34 (1.21, 1.49)
110~119	7698 (45.5)	806 (44.6)	Ref.	Ref.	1012 (42.4)	Ref.	Ref.
120~129	3007 (17.8)	429 (23.8)	1.33 (1.17, 1.51)	1.40 (1.23, 1.59)	396 (16.6)	0.99 (0.87, 1.12)	0.87 (0.76, 0.99)
130~139	458 (2.7)	60 (3.3)	1.19 (0.89, 1.58)	1.28 (0.96, 1.70)	61 (2.6)	0.98 (0.74, 1.30)	0.88 (0.66, 1.17)
≥140	94 (0.6)	11 (0.6)	1.02 (0.53, 1.94)	1.04 (0.54, 1.99)	18 (0.8)	1.53 (0.91, 2.56)	1.33 (0.77, 2.30)
Late-third trimester							
<100	529 (2.8)	49 (2.6)	0.98 (0.72, 1.33)	0.92 (0.67, 1.25)	91 (3.5)	1.25 (0.98, 1.58)	1.38 (1.09, 1.76)
100~109	4541 (24.3)	375 (19.8)	0.86 (0.76, 0.98)	0.83 (0.73, 0.94)	741 (28.5)	1.23 (1.11, 1.36)	1.29 (1.16, 1.44)
110~119	8534 (45.8)	819 (43.3)	Ref.	Ref.	1121 (43.2)	Ref.	Ref.
120~129	4080 (21.9)	496 (26.2)	1.26 (1.12, 1.42)	1.31 (1.16, 1.48)	503 (19.4)	0.94 (0.83, 1.05)	0.85 (0.76, 0.96)
130~139	849 (4.6)	131 (6.9)	1.58 (1.30, 1.94)	1.78 (1.45, 2.19)	119 (4.6)	1.06 (0.86, 1.30)	0.90 (0.73, 1.11)
≥140	119 (0.6)	22 (1.2)	1.84 (1.14, 2.96)	1.96 (1.20, 3.18)	21 (0.8)	1.35 (0.84, 2.17)	1.16 (0.71, 1.91)

Lightweight, small for gestational age (SGA) or low birth weight (LBW); Heavyweight, large for gestational age (LGA) or macrosomia. † The model was adjusted for maternal age, education, parity, gestational age of hemoglobin measurement, neonatal gender, gestational age at delivery and mode of delivery. ‡ The model was additionally adjusted for maternal BMI in the first trimester and weight gain during pregnancy.

**Table 3 nutrients-14-02542-t003:** Associations between change of maternal hemoglobin concentrations and adverse birth outcomes.

ΔHb	Normalweight	Lightweight			Heavyweight		
	N (%)	N (%)	Model 1 †	Model 2 ‡	N (%)	Model 1 †	Model 2 ‡
			OR (95% CI)	OR (95% CI)		OR (95% CI)	OR (95% CI)
Hbfs							
Q1 (−50~−17)	4332 (25.2)	444 (24.4)	Ref.	Ref.	724 (30.2)	Ref.	Ref.
Q2 (−17~−11)	4402 (25.6)	429 (23.6)	0.94 (0.82, 1.09)	0.98 (0.85, 1.14)	642 (26.8)	0.90 (0.80, 1.02)	0.83 (0.73, 0.94)
Q3 (−11~−5)	4407 (25.7)	476 (26.2)	1.07 (0.93, 1.25)	1.17 (1.00, 1.36)	537 (22.4)	0.75 (0.66, 0.86)	0.65 (0.57, 0.75)
Q4 (−5~50)	4034 (23.5)	467 (25.7)	1.16 (0.98, 1.38)	1.32 (1.11, 1.57)	495 (20.6)	0.75 (0.65, 0.88)	0.62 (0.53, 0.72)
Hbft							
Q1 (−50~−16)	4906 (26.3)	391 (20.7)	Ref.	Ref.	831 (32.1)	Ref.	Ref.
Q2 (−16~−9)	4572 (24.6)	438 (23.2)	1.29 (1.11, 1.50)	1.32 (1.14, 1.54)	626 (24.2)	0.82 (0.73, 0.93)	0.78 (0.69, 0.89)
Q3 (−9~−2)	4527 (24.3)	477 (25.2)	1.45 (1.24, 1.69)	1.54 (1.32, 1.80)	579 (22.3)	0.78 (0.69, 0.89)	0.71 (0.63, 0.82)
Q4 (−2~50)	4617 (24.8)	584 (30.9)	1.81 (1.53, 2.14)	1.97 (1.66, 2.33)	555 (21.4)	0.75 (0.65, 0.87)	0.65 (0.56, 0.76)
Hbst *							
Q1 (−50~−3.5)	4362 (26.3)	347 (20.7)	Ref.	Ref.	678 (29.5)	Ref.	Ref.
Q2 (−3.5~2)	4477 (27.0)	431 (25.7)	1.18 (1.01, 1.36)	1.17 (1.01, 1.36)	610 (26.6)	0.91 (0.81, 1.03)	0.93 (0.82, 1.05)
Q3 (2~7)	3727 (22.5)	405 (24.1)	1.34 (1.15, 1.56)	1.33 (1.14, 1.55)	454 (19.8)	0.82 (0.72, 0.93)	0.82 (0.72, 1.01)
Q4 (7~50)	4000 (24.1)	497 (29.6)	1.49 (1.29, 1.73)	1.45 (1.25, 1.68)	555 (24.2)	0.96 (0.85, 1.09)	1.00 (0.88, 1.14)

Lightweight, small for gestational age (SGA) or low birth weight (LBW); Heavyweight, large for gestational age (LGA) or macrosomia; Hbfs, Hb in late-second minus Hb in first; Hbft, Hb in late-third minus Hb in late-third; Hbst, Hb in late-second minus Hb in late-third; Q, quartile. † The model was adjusted for maternal age, education, parity, weeks of gestation at hemoglobin measurement, neonatal gender, gestational age at delivery, mode of delivery and Hb concentration in the first trimester. ‡ The model was additionally adjusted for maternal BMI in the first trimester and weight gain during pregnancy. * Model 1 was adjusted for maternal age, education, parity, weeks of gestation at hemoglobin measurement, neonatal gender, gestational age at delivery, mode of delivery and Hb concentration in the late-second trimester.

## Data Availability

The data presented in this study are available on request from the corresponding author. The data are not publicly available because they contain information that could compromise the privacy of research participants.

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
