# Peer review of "The Associations of Maternal Hemoglobin Concentration in Different Time Points and Its Changes during Pregnancy with Birth Weight Outcomes"

_nutrients, 2022, doi:10.3390/nu14122542_

Round 1

Reviewer 1 Report

Authors aim to investigate the potential association of trimester specific Hb concentrations and changes during pregnancy with birth weight. Although they have a large number of participants and they have done several detailed analyses, I am not sure which is the gap in existing knowledge and what is the novel with this study. Moreover, authors made several assumptions in their analyses eg. they combined different definitions for defining lightweight and heavyweight newborns which does not seem to be accurate.

Comment 1. Introduction. The first paragraph is too long and not focused to the specific research question.

Comment 2. Introduction, general comment: Authors do not clearly and sufficiently explain the rationale for assessing the association of Hb concentration with birth weight. Moreover, there is not good presentation of research findings from other studies and what is the gap in knowledge so far. Also which is their hypothesis behind assessing trimester specific associations? Do they expect that Hb concentrations in a particular trimester are important and why?

Comment 3: title: neonatal birth weight related outcomesà just birth weight, you do not need neonatal.  Authors only look on birth weight either as continuous or categorical. There are no other related outcomes examined. Please correct this in the text as well.

Comment 4: . Page 2, introduction, paragraph 2: There are many ambiguities such as : “Both restricted growth and excessive growth were derived from altered metabolism in the uterus [13], which was closely related to maternal nutritional status”, “Another study found that the lowest maternal Hb during pregnancy might have an inverted U-shaped relationship with neonatal birth weight” “For the adverse outcomes..” (?), “and then reaches the peak at about 34–36 gestational weeks, after that it keeps the peak or slowly return to level at 10 weeks of gestational age”, “In addition, the ability of placental 88 to adapt the nutrient status of the uterus, which was thought to maximize fetal growth 89 and viability at birth, also depended on the gestational timing”

Comment 5: Methods. Is the EMRS a national system? . Zhoushan Maternal and Child Care Hospital is a non-private sector hospital?

Comment 6: methods. What do you mean by “follow up period” ?

Comment 7: why did you use this inclusion  criterion? pregnant women had at least one record of hemoglobin concentration during each period 

Comment 8: why did you choose this criterion ? pregnant women delivered baby after 32 weeks of gestational age . why not restricted only to full-term?

Comment 9: Please explain how did you choose these five time periods. first (5-13 weeks of gestational age), early-second (14-17 weeks of gestational age), mediate-second (18-22 weeks of gestational age), late-second (23-27 weeks of gestational age), early-third (28-31 weeks of gestational age), late-third (from 32 weeks of gestational age to the time of delivery). 

Comment 10: This sentence it is not clear to me. “Meanwhile, the mean Hb concentration was classified into 6 categories by an interval of 138 10 g/L: <100 g/L, 100 ~ 109 g/L, 110 ~ 119 g/L, 120 ~ 129 g/L, 130 ~ 139 g/L, ≥140 g/L “

Comment 11: PAGES 3-4, Lines 141-151 are not clear at all. 

Comment 12: why did you use this exclusion criterion “maternal BMI at first trimester >40 kg/m2 or < 15 kg/m2; “

Comment 13: Why did you combine definitions? A preterm baby could be LBW but not SGA and this is normal. I do not think you should combine these definitions but rather examine them as different outcomes taking into account what is the interpretation of each definition. Moreover, which is the reference percentiles for SGA and LGA definition?

Comment 14: How did you choose for which covariates to adjust for?

Comment 15: Methods. Any sensitivity analyses? E.g. excluding preterm births? Interaction of associations with gender? 

Comment 16: Results. Authors should inform the readers how many were excluded due to the criterion applied and particularly for haw many pregnancies there were missing data on exposure and/or outcome respectively. Briefly assess if there are any differences between participants and non-participants.

Comment 17: Table 1. apart form mean gestational age it is interesting to provide numbers(%) of births between 32-37 weeks and then >37 weeks.

Comment 18: Results in table 2 could be better presented as a figure e.g. box plot

Comment 19: Line 207. Which is Figure A1?

Comment 20: Figure 2 and table 3. As I have already stated, I don’t think it is a good idea to combine definitions.

Comment 21: Discussion. Difficult to follow the because there is a lot of repetition of the observed findings and the results from many other studies.

Comment 22: Discussion. Hb is a proxy of other situations, pathological changes occurring in pregnancy indicating potential mechanisms for the observed associations. Authors refer to one of them e.g. plasma volume expansion. Authors do not explain other potential mechanisms for the observed associations.

Comment 23: Discussion. The generalizability of the research findings and their importance in clinical practice are not discussed.

Author Response

Comment 1. Introduction. The first paragraph is too long and not focused to the specific research question.

Response 1: Thanks for your comment. It has been revised in the manuscript, line 39-53.

Comment 2. Introduction, general comment: Authors do not clearly and sufficiently explain the rationale for assessing the association of Hb concentration with birth weight. Moreover, there is not good presentation of research findings from other studies and what is the gap in knowledge so far. Also which is their hypothesis behind assessing trimester specific associations? Do they expect that Hb concentrations in a particular trimester are important and why?

Response 2: It has been revised in the manuscript. Most studies that evaluating the relationship between maternal Hb in different trimesters and birth weight agreed that Hb concentrations in a particular trimester are important because maternal Hb fluctuated and placental adaptability during pregnancy.

However, the conclusions of different studies were still conflicting. We speculated that the contradiction might due to inaccurate definition of gestational age for maternal Hb measurements. Therefore, we evaluated maternal associations between Hb at different time-points and birth weight outcomes, moreover, the maternal Hb concentration changes between specific time were also assessed in this study.

Comment 3: title: neonatal birth weight related outcomes just birth weight, you do not need neonatal.  Authors only look on birth weight either as continuous or categorical. There are no other related outcomes examined. Please correct this in the text as well.

Response 3: It has been revised through the whole manuscript.

Comment 4: . Page 2, introduction, paragraph 2: There are many ambiguities such as : “Both restricted growth and excessive growth were derived from altered metabolism in the uterus [13], which was closely related to maternal nutritional status”, “Another study found that the lowest maternal Hb during pregnancy might have an inverted U-shaped relationship with neonatal birth weight” “For the adverse outcomes..” (?), “and then reaches the peak at about 34–36 gestational weeks, after that it keeps the peak or slowly return to level at 10 weeks of gestational age”, “In addition, the ability of placental 88 to adapt the nutrient status of the uterus, which was thought to maximize fetal growth 89 and viability at birth, also depended on the gestational timing”

Response 4: Sorry for the ambiguities. It has been revised in the manuscript:

“Both restricted growth and excessive growth of neonates were derived from altered metabolism in the uterus [12], which was closely related to maternal nutritional status.”

“Another study measured maternal Hb concentration at least once during pregnancy, and found that the lowest maternal Hb during pregnancy might have an inverted U-shaped relationship with neonatal birth weight [16].”

“Furthermore”

“and then reaches the peak at about 34–36 gestational weeks, after that it keeps the peak or slowly return to level that at 10 weeks of gestational age”

“In addition, the ability of placental to adapt the uterine nutrition, which was thought to maximize fetal growth and viability at birth, also depended on the gestational timing”

Comment 5: Methods. Is the EMRS a national system? . Zhoushan Maternal and Child Care Hospital is a non-private sector hospital?

Response 5: EMRS is a municipal system, and Zhoushan Maternal and Child Care Hospital is a non-private sector hospital. It has been revised in the manuscript, line 106-107.

Comment 6: methods. What do you mean by “follow up period” ?

Response 6: It has been revised in the manuscript: “follow up date”, line 115. The “follow up date” mean the date when a pregnant woman had an antenatal visit.

Comment 7: why did you use this inclusion  criterion? Pregnant women had at least one record of hemoglobin concentration during each period 

Response 7: Your suggestion is reasonable, so the inclusion criteria has been revised to “pregnant women had at least one record of hemoglobin concentration during pregnancy”, then all results in this study has been updated and the main findings are very stable. Moreover, the subgroup analysis was performed by using the population in which pregnant women had at least one record of Hb concentration during each defined period (Table A7 and Table A8).

Comment 8: why did you choose this criterion ? pregnant women delivered baby after 32 weeks of gestational age . why not restricted only to full-term?

Response 8: The association between maternal Hb and LBW is one of study aims, and most of infants with LBW were delivery before 37 weeks of gestational age; but the sensitivity analysis conducted in pregnant women who delivered baby after 37 weeks of gestational age (Table A5 and Table A6).

Comment 9: Please explain how did you choose these five time periods. first (5-13 weeks of gestational age), early-second (14-17 weeks of gestational age), mediate-second (18-22 weeks of gestational age), late-second (23-27 weeks of gestational age), early-third (28-31 weeks of gestational age), late-third (from 32 weeks of gestational age to the time of delivery). 

Response 9: The pregnant women usually received the first prenatal check-up between 5 and 14 weeks of gestational age and then took check-ups every four weeks in the second trimester, every two weeks before 37 weeks of gestational age. However, due to our data limitation, mostly pregnant women had 2 0r 3 Hb records in the third trimester. In addition, according to the existing knowledge, maternal Hb concentration reached the minimal levels at 34-36th week of gestational age and changed a little thereafter. Therefore, we selected the five timepoints to examine the association between Hb concentration and birth weight outcomes.

Comment 10: This sentence it is not clear to me. “Meanwhile, the mean Hb concentration was classified into 6 categories by an interval of 138 10 g/L: <100 g/L, 100 ~ 109 g/L, 110 ~ 119 g/L, 120 ~ 129 g/L, 130 ~ 139 g/L, ≥140 g/L “

Response 10: Thanks. It has been revised as: “Meanwhile, the mean Hb concentration that between 100 and 140 g/L was classified into 6 categories by an interval of 10 g/L: <100 g/L, 100 ~ 109 g/L, 110 ~ 119 g/L, 120 ~ 129 g/L, 130 ~ 139 g/L, ≥140 g/L.”

Comment 11: PAGES 3-4, Lines 141-151 are not clear at all. 

Response 11: It has been revised in the manuscript.

The main purpose of this paragraph was to explain how we choose the maternal Hb concentration in different trimesters to reflect the fluctuation in Hb during pregnancy.

Comment 12: why did you use this exclusion criterion “maternal BMI at first trimester >40 kg/m2 or < 15 kg/m2; “

Response 12: The values of maternal early-pregnancy BMI >40 kg/m2 or < 15 kg/m2 were extreme data in this population.

Comment 13: Why did you combine definitions? A preterm baby could be LBW but not SGA and this is normal. I do not think you should combine these definitions but rather examine them as different outcomes taking into account what is the interpretation of each definition. Moreover, which is the reference percentiles for SGA and LGA definition?

Response 13: Small for gestational age (SGA) and low birth weight (LBW) represented a status of growth restriction for newborns, large for gestational age (LGA) and macrosomia represented a status of excessive growth for newborns, SGA and LGA were birth weight outcomes defined relative to neonates of the same gestational age and gender, while LBW and macrosomia were defined on an absolute level, both of which related to birth weight-related status. Therefore, we considered SGA and LBW as the similar pathological condition and combined the definitions as compound outcomes, the same was true for SGA and macrosomia. Moreover, the similar association of Hb concentrations with SGA and LBW was found.

The reference percentiles for SGA and LGA definition were 10th and 90th, respectively. (line 154)

Comment 14: How did you choose for which covariates to adjust for?

Response 14: We adjusted for the general socio-demographic characteristics that might contribute to confounding bias such as maternal age, maternal education, parity, neonatal gender, which had also been adjusted for in other studies when birth weight was used as the outcomes [1,2]. Additionally, we adjusted for the variables that were significantly different between groups, such as gestational age at delivery, mode of delivery, maternal early-pregnancy BMI and weight gain during pregnancy. Moreover, the association of maternal Hb with birth weight outcomes was affected by gestation age of hemoglobin measurement [3].

Comment 15: Methods. Any sensitivity analyses? E.g. excluding preterm births? Interaction of associations with gender? 

Response 15: Thanks for your suggestion. We tested the interaction between gender Hb, but didn't find the significant interaction. In addition, we conducted a sensitivity analysis after excluding preterm births and observed similar results as those in the full population. The results were presented in the supplementary materials, Table A5 and Table A6.

Comment 16: Results. Authors should inform the readers how many were excluded due to the criterion applied and particularly for haw many pregnancies there were missing data on exposure and/or outcome respectively. Briefly assess if there are any differences between participants and non-participants.

 Response 16: Thanks for your suggestion. The flow chart of participants and the comparison of general information between participants and non-participants were added in the supplementary materials, Figure A1 and Table A1.

Comment 17: Table 1. apart form mean gestational age it is interesting to provide numbers(%) of births between 32-37 weeks and then >37 weeks.

Response 17: It had been presented in Table 1. The variable “preterm” mean birth between 32-37 weeks of gestational age.

Comment 18: Results in table 2 could be better presented as a figure e.g. box plot

Response 18: Based on your suggestion, a figure was generated:

Figure 1 Maternal hemoglobin concentration(a) and percentage of maternal hemoglobin status(b) stratified by birth weight outcomes in differet time points of pregnancy. f, first trimester; s1, early-second trimester; s2, mediate-second trimester; s3, late-second trimester; t1, early-third trimester; t2, late-third trimester

Comment 19: Line 207. Which is Figure A1?

Response 19: Sorry for the mistake. After updating the result, it has been revised in the manuscript: “The no-linear association of maternal Hb in the early-second and mediate-second trimester with neonatal birth weight were not observed (Figure A2)”. 

Comment 20: Figure 2 and table 3. As I have already stated, I don’t think it is a good idea to combine definitions.

Response 20:  Please see response 13

Comment 21: Discussion. Difficult to follow the because there is a lot of repetition of the observed findings and the results from many other studies.

Response 21: Sorry for that. It has been revised in the manuscript.

Comment 22: Discussion. Hb is a proxy of other situations, pathological changes occurring in pregnancy indicating potential mechanisms for the observed associations. Authors refer to one of them e.g. plasma volume expansion. Authors do not explain other potential mechanisms for the observed associations.

Response 22: Thanks for your comment. It has been revised in the manuscript.

Comment 23: Discussion. The generalizability of the research findings and their importance in clinical practice are not discussed.

Response 23: The generalizability of the research findings might be limited due to the significant differences between participants and non-participants. We discussed it in the part 5, line 377-380. However, subgroup analysis and sensitivity analysis were conducted to examine the stability of the results.

The important in clinical practice has been discussed in the manuscript, line 335-360.

References

  1. Wang, X.; Zuckerman, B.; Coffman, G.A.; Corwin, M.J. Familial aggregation of low birth weight among whites and blacks in the United States. The New England journal of medicine 1995, 333, 1744-1749, doi:10.1056/nejm199512283332606.
  2. Smith, G.C.; Smith, M.F.; McNay, M.B.; Fleming, J.E. First-trimester growth and the risk of low birth weight. The New England journal of medicine 1998, 339, 1817-1822, doi:10.1056/nejm199812173392504.
  3. Bakacak, M.; Avci, F.; Ercan, O.; Köstü, B.; Serin, S.; Kiran, G.; Bostanci, M.S.; Bakacak, Z. The effect of maternal hemoglobin concentration on fetal birth weight according to trimesters. The journal of maternal-fetal & neonatal medicine : the official journal of the European Association of Perinatal Medicine, the Federation of Asia and Oceania Perinatal Societies, the International Society of Perinatal Obstet 2015, 28, 2106-2110, doi:10.3109/14767058.2014.979149.

Reviewer 2 Report

Dear Authors,

    I have had the opportunity to read your manuscript entitled “The associations of maternal hemoglobin concentration in different time points and it changes during pregnancy with neonatal birth weight-related outcomes”, which shows an exhaustive analysis of the complex interaction between maternal hemoglobin throughout the three trimesters of gestation and fetal growth/birth weight.

A meticulous analysis of the variations of hemoglobin has been developed thanks to a division of second and third trimesters into well-established time periods, to better identify the most relevant time-points of influence on birth weight. You have specifically classified the data of birth outcomes as compound outcomes which clarifies the potential associations.

The tables and figures are very clear and comprehensive. The discussion is well written and contains the pertinent references to address the findings of this study.

The manuscript is remarkable; however, there are some minor aspects that they should be highlighted:

-        There is no information at all about the use of iron supplements. This point must be presented as a limitation of the study.

-        Other limitations: you could not ascertain the role of other maternal conditions (hypertensive disorders, gestational diabetes, or placental insufficiency) that might significantly interfere the associations between hemoglobin and birth weight. It would be desirable to include these aspects at the discussion.

-        Lines 286-288:  This sentence doesn’t contain a verb; please rephrase it.

Author Response

-        There is no information at all about the use of iron supplements. This point must be presented as a limitation of the study.

Response: It has been revised in section 5, line 366-367.

-        Other limitations: you could not ascertain the role of other maternal conditions (hypertensive disorders, gestational diabetes, or placental insufficiency) that might significantly interfere the associations between hemoglobin and birth weight. It would be desirable to include these aspects at the discussion.

Response: Considering that some gestational diseases may have affected the results, we excluded pregnant women with GDM and hypertensive disorders on the available data, which was mentioned in the exclusion criteria in Methods, line 126-127. However, it is regrettable that the information about other diseases such as placental insufficiency is not available. The discussion of this limitation has been added in part 5, line 373-375.

-        Lines 286-288:  This sentence doesn’t contain a verb; please rephrase it.

Response: Thank you for your reminder. I rephrased it in the article, line 285-288.